# Effects of Durum Wheat Cultivars with Different Degrees of FHB Susceptibility Grown under Different Meteorological Conditions on the Contamination of Regulated, Modified and Emerging Mycotoxins

**DOI:** 10.3390/microorganisms9020408

**Published:** 2021-02-16

**Authors:** Valentina Scarpino, Massimo Blandino

**Affiliations:** Department of Agricultural Forest and Food Sciences, University of Turin, Largo Paolo Braccini 2, 10095 Grugliasco, Italy; valentina.scarpino@unito.it

**Keywords:** 3-acetyldeoxynivalenol, deoxynivalenol, deoxynivalenol-3-glucoside, enniatins, moniliformin, zearalenone

## Abstract

The enhancement of *Fusarium* head blight (FHB) resistance is one of the best options to reduce mycotoxin contamination in wheat. This study has aimed to verify that the genotypes with high tolerance to deoxynivalenol could guarantee an overall minimization of the sanitary risk, by evaluating the contamination of regulated, modified and emerging mycotoxins on durum wheat *cvs* with different degrees of FHB susceptibility, grown under different meteorological conditions, in 8 growing seasons in North-West Italy. The years which were characterized by frequent and heavy rainfall in spring were also those with the highest contamination of deoxynivalenol, zearalenone, moniliformin, and enniatins. The most FHB resistant genotypes resulted in the lowest contamination of all the mycotoxins but showed the highest deoxynivalenol-3-glucoside/deoxynivalenol ratio and moniliformin/deoxynivalenol ratio. An inverse relationship between the amount of deoxynivalenol and the deoxynivalenol-3-glucoside/deoxynivalenol ratio was recorded for all the *cvs* and all the years. Conversely, the enniatins/deoxynivalenol ratio had a less intense relationship with *cv* tolerance to FHB. In conclusion, even though the more tolerant *cvs*, showed higher relative relationships between modified/emerging mycotoxins and native/target mycotoxins than the susceptible ones, they showed lower absolute levels of contamination of both emerging and modified mycotoxins.

## 1. Introduction

Crop productions will be affected by global warming at various rates in different parts of the world, particularly in the most critical and vulnerable geographic areas, like the Mediterranean Basin [1]. Despite its relatively high adaptability to marginal and drought-affected environments, durum wheat could be negatively influenced by the impacts of climate change and is highly sensitive to climatic and environmental variations [2]. 

Furthermore, as a result of global warming (i.e., less severe winter temperatures and more severe dry periods during summer, which could limit the cultivation of summer crops), the geographic distribution of winter wheat might change, especially toward northern latitudes, and it might adapt better to these growing areas [3]. Moreover, in some countries, such as Italy, the pasta industry is looking for national durum wheat productions with high traceability and quality levels, as far as a high grain protein content is concerned. These qualitative targets can be achieved through the cultivation of winter durum wheat in more temperate and fertile growing areas, such as the Po Valley in Northern Italy. However, durum wheat in temperate areas is characterized by a generally higher susceptibility to fungal diseases than the common one, thus limiting its potential cultivation in these areas [4]. 

In addition to the direct effects on crop species [5], climate change may act indirectly on the security and quality of food by altering interactions with pests [6,7], as well as impacting the disease triangle involving host, pathogen and environment [8]. 

One of the most serious biotic threats is related to the fact that *Fusarium* head blight (FHB) is able to affect both the yield potential and the sanitary risk of wheat to a great extent [9]. Moreover, among the most important hazards likely to be affected by climate change, and which is linked to FHB, is contamination by mycotoxins [10]. These compounds are secondary metabolites, which are harmful to both humans and animals, that are produced by filamentous fungi. Deoxynivalenol (DON), a type-B trichothecene produced mainly by *F. graminearum sensu stricto* and *F. culmorum*, is the most prevalent toxin in small cereal crops throughout the world and in Europe [11]. 

Regulatory limits have been set by the European Commission (EC) to protect humans from exposure to this mycotoxin through cereal consumption (EC Regulation No. 1881/2006, with a limit of 1750 µg kg^−1^ in unprocessed durum wheat) [12]. However, recent quantitative estimations have shown that increased DON contamination can be expected in cereals in certain regions in Europe as a result of future climate change [13,14]. Although the contamination of wheat grains by DON depends mainly on the meteorological conditions, particularly at flowering [15], an important role is also played by agronomic factors [16]. 

At present, the most effective approach adopted to minimize the occurrence of DON in wheat is the use of preventive agronomic practices to reduce the pathogen infection and development, and the accumulation of mycotoxins in grains. Thus, an increased tolerance to *Fusarium* infection and mycotoxin contamination should remain a priority for the genetic improvement of wheat *cvs*. Although breeding programs are hindered by the fact that the resistance to FHB is under polygenic inheritance, the efforts to genetically improve of durum wheat have recently been directed toward releasing *cvs* on the market that are more tolerant to FHB [17] and thus which have been designed for cultivation in cropping systems in temperate growing areas. Furthermore, the selection of genotypes that are less prone to FHB is generally made by only focusing on DON, as it is the target mycotoxin currently considered by supply chains and in the regulatory limits. 

To the best of our knowledge, few research efforts have been made to investigate the influence of durum wheat genotypes, and their interaction with environmental conditions, on the occurrence and the relative amount of emerging and modified mycotoxins in grains. 

Emerging mycotoxins are commonly defined as “mycotoxins which are neither routinely determined nor legislatively regulated; however, the evidence of their incidence is rapidly increasing” [11]. Among the regulated mycotoxins, in addition to DON, also T-2 and HT-2 toxins, zearalenone and ochratoxin A (OTA) could be found to contaminate durum wheat in field, while the occurrence of aflatoxins (AFs) is infrequent [18]. 

On the other hand, ergot alkaloids, enniatins (ENNs), beauvericin (BEA) and moniliformin (MON) are those that are more commonly mentioned in the emerging mycotoxin group [19]. 

EFSA, the European Authority for the Safety of Food, has recently been paying more attention to this group of mycotoxins, and risk assessment studies are still in progress. The term “modified mycotoxins” was introduced by Rychlik et al. in 2014 [20] to refer to any mycotoxin whose structure has been changed during a chemical/biochemical reaction [21]. 

Plants play a key role in biologically modifying mycotoxins as a primary plant mechanism against fungal infection and mycotoxin accumulation, producing the so-called “masked mycotoxins”. As far as DON is concerned, the conjugation of DON with glucose (DON-3-glucoside, DON-3-G) is the wheat mechanism (phase II metabolism) that is involved the most in the resistance to DON accumulation and it is closely linked to the wheat genotypes [22].

Nowadays, both emerging and modified mycotoxins are receiving increasing attention by the scientific community, and by governments and regulators, due to the presence of some of them in high concentrations in food and feeds, and because of their toxic effect [23]. The drafting of legislation that will set limits for some of these mycotoxins is expected in the near future. 

Since the occurrence of masked mycotoxins is a strategy that is implemented by a plant to limit the accumulation of their free native forms, it is advisable to verify the effect of *cv* selection considering the total risk associated with these toxic compounds. Moreover, since emerging mycotoxins, such as ENNs and MON, could be produced by other *Fusarium* species than those primary responsible for DON occurrence, more detailed knowledge on the environmental and agronomic conditions that promote their occurrence is essential, in order to set up field programs that would be able to minimize the overall sanitary risk. 

Therefore, the aim of the present research has been to evaluate the effects of durum wheat *cvs* with different degrees of FHB susceptibility and the meteorological conditions that are able to lead to different degrees of disease pressure on the contamination of regulated, modified and emerging mycotoxins in 8 growing seasons in North-West Italy. 

The practical objective, in a food supply chain framework, is to verify that the genotypes selected because of their high tolerance to DON are overall able to guarantee a minimization of the sanitary risk, according to a holistic approach that also considers the modified DON forms and the emerging mycotoxins.

## 2. Materials and Methods

### 2.1. Design of the Field Experiment and Samples

The effects of the choice of a durum wheat cultivar (*cv*) and the meteorological conditions on the contamination of regulated, modified and emerging mycotoxins was studied in North-West Italy over a period of 8 years at Cigliano (45° 18′ N, 8° 01′ E; altitude 237 m), on plants grown in a sandy-loam soil (Typic Hapludalfs).

This growing area is characterized by a Cfa climate, that is, a humid subtropical climate according to the Köppen climate classification [24] and by a probable high FHB pressure, due to the environmental and agronomic conditions (frequent rotation with maize).

The daily temperatures and precipitation were measured at a meteorological station near the experimental area. Two durum wheat *cvs* with different degrees of susceptibility to DON contamination, that is, Saragolla (classified as susceptible) and SY Cysco (classified as moderately tolerant), were compared each year under naturally-infected field conditions.

Secolo, Odisseo and Fuego *cvs*, which are characterized by an intermediate susceptibility to DON contamination, were also compared in the same experimental field, albeit only over four years (2016–2019). All the *cvs* were provided by Syngenta Italia Spa Milano, Italy.

The agronomic growing technique commonly adopted in the area was applied. Briefly, the previous crop was maize, the field was ploughed each year, incorporating the debris into the soil, and this was followed by disk harrowing to prepare a suitable seedbed. Planting was conducted in 12 cm wide rows at a seeding rate of 450 seeds m^−2^ in October or November.

A total of 170 kg N ha^−1^ was applied, 130 at wheat tillering [growth stage (GS) 23] [25] as ammonium sulfate-nitrate with nitrification inhibitors and 40 kg N ha^−1^ at heading (GS 51) as an ammonium nitrate fertilizer.

A strobilurin fungicide (azoxystrobin active ingredient, applied at 0.25 kg ha^−1^, produced by Amistar^®^, Syngenta Italia Spa, formulation: suspension concentrate) was applied at the booting stage (GS 45) to control foliar diseases. No fungicide was applied at flowering (GS61-65) to control FHB infection. The sowing and harvest dates are reported in Table 1, together with the dates of the main growth stages, for each growing season.

Treatments were assigned to an experimental unit using a completely randomized block design with three replicates. The plots measured 7 × 1.5 m^2^.

The grain yields were obtained by harvesting the whole plot using a Walter Wintersteiger cereal plot combine-harvester. A subsample was taken from each plot to determine the grain moisture and the test weight (TW). The TW was determined using a Dickey-John GAC2000 grain analysis meter, according to the supplied program.

The grain yield results were adjusted to a 13% moisture content. The harvested grains were mixed thoroughly, and 4 kg grain samples were taken from each plot, ground using a ZM 200 Ultra Centrifugal Mill (Retsch GmbH, Haan, Germany) and representative sub-samples of each flour were used directly to analyze the mycotoxin content.

### 2.2. FHB Symptoms

FHB incidence and severity were recorded for each plot, by carrying out visual evaluations of the disease at the soft dough stage (GS 85).

FHB head blight incidence was calculated as the percentage of ears with symptoms when 200 ears per plot were analyzed.

FHB severity was calculated as the percentage of kernels per ear with symptoms.

A scale of 1 to 7 was used in which each numerical value corresponded to a percentage interval of surfaces exhibiting visible symptoms of the disease, according to the following schedule: 1 = 0–5%, 2 = 6–15%, 3 = 16–30%; 4 = 31–50%, 5 = 51–75%, 6 = 76–90%, 7 = 91–100% [26].

The FHB severity scores were converted into percentages of the ear exhibiting symptoms, replacing each score with the mid-point of the interval.

### 2.3. Multi-Mycotoxin LC-MS/MS Analysis

#### 2.3.1. Extraction and Sample Preparation

The extraction and sample preparation were performed according to the dilute-and-shoot method reported by Scarpino et al. in 2019 [27].

Briefly, 5 g of wheat flour was extracted, by means of mechanical shaking, with 20 mL of CH_3_CN/H_2_O/CH_3_COOH (79/20/1, v/v/v).

The filtered extract was diluted with the same volume of CH_3_CN/H_2_O/CH_3_COOH 20/79/1, *v*/*v*/*v*, vortexed and filtered again through 15 mm diameter, 0.2 µm regenerated cellulose (RC) syringe filters (Phenex-RC, Phenomenex, Torrance, CA, USA) and 20 µL of the diluted filtered extract was analyzed without any further pre-treatment.

#### 2.3.2. LC-MS/MS Analysis

LC-MS/MS analysis was carried out on a Varian 310 triple quadrupole (TQ) mass spectrometer (Varian, Italy), equipped with an electrospray ionization (ESI) source, a 212 LC pump, a ProStar 410 AutoSampler and dedicated software.

Liquid chromatography (LC) separation was performed on a Gemini-NX C_18_ 100 × 2.0 mm i.d., 3 μm particle size, 110 Å, equipped with a C_18_ 4 × 2 mm security guard cartridge column (Phenomenex, Torrance, CA, USA), using water (eluent A) and methanol (eluent B), both acidified with 0.1% *v*/*v* CH_3_COOH, as eluents that were delivered at 200 µL min^−1^. The chromatographic and mass spectrometric conditions were described in detail by Scarpino et al. [27].

The results pertaining to the linearity range, the limit of detection (LOD), the limit of quantification (LOQ), the apparent recovery R_A_ (%), the matrix effects obtained through the evaluation of the signal suppression/enhancement SSE (%) and the recovery of the extraction R_E_ (%) were reported by Scarpino et al. [27].

### 2.4. Statistical Analysis

The normal distribution and homogeneity of variances were verified by performing the Kolmogorov–Smirnov normality test and the Levene test, respectively.

An analysis of variance (ANOVA) was utilized to compare grain yield, TW, FHB incidence and severity, using a completely randomized block design, in which the year and the durum wheat *cvs,* with different degrees of FHB susceptibility, were the independent variables.

Moreover, ANOVA was conducted separately for each year to evaluate the effect of the durum wheat *cvs,* with different FHB susceptibility, on the contamination of regulated, modified and emerging mycotoxins, using a completely randomized block design. In some analyses, the mycotoxin concentrations were transformed, using the y’ = ln(x + 1) equation, to normalize the residuals.

Simple correlation coefficients were obtained for mycotoxins, relative to each other and to meteorological (rainfall and GDD referred to the period November-June, May and June) and agronomic (grain yield and FHB severity) parameters, by joining the data sets that referred to the eight growing seasons.

SPSS for Windows, Version 26.0, (SPSS Inc., Chicago, IL, USA), was used for the statistical analysis.

## 3. Results

### 3.1. Meteorological Data

The 8 growing seasons showed different meteorological trends, as far as both rainfall and temperature (expressed as growing degree days, GDDs) are concerned (Table 2).

Overall, the recent growing seasons showed higher temperatures than the average temperature for the 1990-2010 period, in particular from November to January, and in April and June. On average, the rainfall was also higher than in 1990–2010 (+200 mm), although the trends were highly variable, in terms of total precipitation and their distribution between seasons.

Frequent rainfall (>120 mm) from wheat heading to the end of flowering (May) occurred in 2016, 2018, 2019 and 2020. In 2018 and 2020, the high rainfall in May was combined with high temperatures, while in 2016 and 2019, the GDDs during this period were lower than in the other years. In 2015 and 2017, the limited rainfall and high temperature levels in May and June led to a rapid ripening and an early harvesting time.

### 3.2. Yield Parameters

The recorded grain yields (on average 5.9 t ha^−1^) were in agreement with the yield expected for durum wheat in growing areas without disease control at flowering.

The moderately tolerant SY Cysco *cv* resulted in a significantly higher yield (+6.8%) and TW (+5.8%) than Saragolla (Table 3). The Fuego, Odisseo and Secolo *cv* grain yields did not differ from that of Saragolla, although these varieties all resulted in a significantly higher TW (Table 4). The TW was very low (<77 kg hL^−1^) in 2016 and in 2018.

The interaction between *cv* and year was significant for both the grain yield and TW (Table 3 and Table 4). A greater difference was recorded for the yield parameters between SY Cysco and Saragolla in 2016 (grain yield values of 6.7 and 6.3 t ha^−1^ and TW values of 79.8 and 73.0 kg hL^−1^ for SY Cysco and Saragolla, respectively), 2017 (grain yield values of 6.4 and 5.8 t ha^−1^ and TW values of 80.4 and 75.3 kg hL^−1^ for SY Cysco and Saragolla, respectively) and 2019 (grain yield values of 6.1 and 4.4 t ha^−1^ and TW values of 82.5 and 77.95 kg hL^−1^ for SY Cysco and Saragolla, respectively).

### 3.3. FHB Symptoms

FHB incidence and severity were higher in 2016, followed by 2020, 2019 and 2018 (Table 3).

The medium tolerant *cv* SY Cysco resulted in a significantly lower FHB incidence (–44.1%) and severity (–10.7%) than the susceptible Saragolla *cv*.

The Secolo, Odisseo and Fuego *cvs* confirmed intermediate behavior between Saragolla and SY Cysco, as far as the FHB symptoms are concerned (Table 4).

On average, Fuego did not differ from SY Cysco, while Secolo resulted in a higher amount of disease symptoms.

The interaction between *cv* and year was always significant. The differences between susceptible *cv* Saragolla and SY Cysco were higher in 2016 (FHB incidence values of 98.7 and 48.4% and FHB severity values of 36.7 and 5.0% for Saragolla and SY Cysco, respectively), 2018 (FHB incidence values of 71.3 and 18.1% and FHB severity values of 19.8 and 0.6% for Saragolla and SY Cysco, respectively) and 2019 (FHB incidence values of 75.0 and 32.2% and FHB severity values of 11.7 and 1.8% for Saragolla and SY Cysco, respectively).

### 3.4. Mycotoxin Contamination

About 10 mycotoxins were detected at the same time in at least one of the analyzed samples: 3-acetyldeoxynivalenol (3-ADON), deoxynivalenol (DON), deoxynivalenol-3-glucoside (DON-3-G), enniatin A (ENN A), enniatin A_1_ (ENN A_1_), enniatin B (ENN B), enniatin B_1_ (ENN B_1_), moniliformin (MON) and zearalenone (ZEA).

DON was detected for all of the years and in all of the *cvs* (Figure 1 and Figure 2). The content of this mycotoxin was clearly related to the meteorological conditions, particularly close to anthesis, in each growing season. The DON contamination for the Saragolla and SY Cysco *cvs* was on average low in 2013, 2015 and 2017 (1888, 1233, 1453 µg kg^−1^, respectively). The average contamination was clearly higher in 2014 (3081 µg kg^−1^), 2016 (5573 µg kg^−1^), 2019 (4540 µg kg^−1^) and 2020 (3856 µg kg^−1^). The highest contamination level was detected in 2018 (10,815 µg kg^−1^). As far as the modified DON forms are concerned, 3-ADON and DON-3-G were on average 5% and 11% of the total DON, respectively. However, 15-ADON was never detected. SY Cysco resulted in a significantly lower contamination of the total DON (−77%) than Saragolla, except for in 2020 (Figure 1).

The DON levels in the Saragolla *cv* were above the current EU regulatory limits (EC Regulation No. 1881/2006, with a limit of 1750 µg kg) [12] for all the considered years. On the other hand, the cultivation of a moderately tolerant *cv*, SY Cysco, allowed the limits to be complied with, without the need of any fungicide application, for half of the considered growing seasons, except in 2016, 2018, 2019 and 2020. Secolo, Odisseo and Fuego resulted in significantly lower levels than Saragolla in all the considered experiments (Figure 2). Their DON content was higher than SY Cysco in 2016, 2017 and 2019, while only in the 2018, with the highest total DON (DON TOT = sum of DON, DON-3-G and 3-ADON) was no difference in contamination observed between these *cvs*. In 2017 and 2019, Fuego showed a significant lower total DON content than Odisseo and Secolo, while Odisseo showed a lower contamination in 2016.

ZEA was detected in 2014, 2016, 2018 and 2019, albeit only in Saragolla, and in both Saragolla and SY Cysco in 2020 (Figure 1). The ZEA contamination in Saragolla was on average 178 µg kg^−1^ in the aforementioned growing seasons. In these years, ANOVA showed a significantly lower ZEA content in SY Cysco than in Saragolla, except in 2019. No difference was recorded between SY Cysco and Secolo, Odisseo and Secolo in these years (Figure 2).

The highest contamination of MON for the Saragolla and SY Cysco *cvs* was detected in 2016 (514 µg kg^−1^), followed by 2015 (251 µg kg^−1^), 2020 (238 µg kg^−1^) and 2018 (231 µg kg^−1^). The content of this emerging mycotoxin was always below 200 µg kg^−1^ in 2013, 2014, 2017 and 2019 (Figure 1). In all the years, SY Cysco showed a significantly lower MON contamination than Saragolla (−57%). Secolo, Odisseo and Fuego also had a significantly lower MON content than Saragolla, except for in 2018, while only in 2017 a significant difference between these *cvs* and SY Cysco was recorded (Figure 2).

ENNs (ENN TOT = the sum of ENN A, ENN A_1_, ENN B, ENN B_1_) was the mycotoxin group with the highest content after DON and its modified forms. On average, ENN A, ENN A_1_, ENN B, ENN B_1_ accounted respectively for 1%, 7%, 71% and 21% of the total enniatin contamination level. The relative rate of each ENN form was quite stable among *cvs* and it varied on average over the years within about ±10–15% of the previous mentioned values. In the same way as for MON, 2016 was the growing season with the highest average contamination of ENN (2287 µg kg^−1^). The high content of these mycotoxins recorded in both 2015 (1002 µg kg^−1^) and in 2020 (1452 µg kg^−1^) highlight the absence of any relationship with the DON levels. The lowest ENN contaminations were observed in 2013 (49 µg kg^−1^) and 2017 (145 µg kg^−1^) (Figure 1). Significant differences between Saragolla and SY Cysco were only observed in 2015 (–62%), 2016 (–86%), 2019 (–43%) and 2020 (–56%), although greater differences than –80% were also recorded in 2013 and 2014. Secolo, Odisseo and Fuego had significantly lower ENN contents than Saragolla for all the considered growing seasons, although they did not differ from SY Cysco, except for a lower contamination of Secolo in 2019 (Figure 2).

### 3.5. Ratio between the Modified and Emerging Mycotoxins and DON

As far as the ratio between the modified mycotoxins and DON is concerned, the DON-3-G/DON molar ratio (MR), was significantly higher for SY Cysco than for Saragolla over all the considered years, except for 2015 and 2020. Thus, SY Cysco showed a two-times higher modifying ability of the parent form of DON than Saragolla (Figure 3). Secolo, Odisseo and Fuego never showed any differences, pertaining to their DON-3-G/DON MR from SY Cysco, except for Fuego in 2016 and Secolo in 2019, when significantly lower ratios were observed (Figure 4). Similarly, SY Cysco had a higher MON/DON ratio than Saragolla in 2013, 2014, 2016, 2017 and 2018, and almost two-times higher values were on average recorded in these years (Figure 3). Secolo, Odisseo and Fuego always showed a significantly lower MON/DON ratio than SY Cysco, except in 2018 (Figure 4). Conversely, the ENN TOT/DON TOT ratio showed variable behavior. Indeed, this ratio was only significantly higher for SY Cysco than for Saragolla in 2017 and in 2019, while it was significantly lower in 2020 (Figure 3). Like Saragolla, the Secolo, Odisseo and Fuego *cvs* also had a significantly lower ENN TOT/DON TOT ratio than SY Cysco in 2017 and 2019. The *cvs* that showed the significantly lowest ENN TOT/DON TOT ratios were Odisseo and Secolo in 2017 and 2019, respectively.

### 3.6. Correlations between Agro-Meteorological Parameters and Mycotoxins

Grain yield significantly and negatively correlated with the rainfall occurred in May and the GDD recorded in May and June, while was positively correlated with the rainfall occurred in June (Table 5). As far as the correlation among mycotoxins, relative to each other and to agro-meteorological parameters was concerned, the occurrence of DON was closely related to the rainfall occurred in May and although less intensely it correlated significantly with ZEA, MON and ENN TOT. DON-3-G exhibited DON-like behavior, with which it was closely related. On the other hand, the DON-3-G/DON ratio was negatively related to the DON content. ZEA, MON and ENN TOT were closely related to each other, and negatively related to the GDD recorded in June, while, except for MON, they were not influenced by the rainfall occurred in May.

## 4. Discussion

The data collected in the present research highlight, for the first time ever, co-contamination by regulated (DON and ZEA), modified (DON-3-G and 3-ADON) and emerging *Fusarium* mycotoxins (MON and ENN A, A_1_, B, B_1_) of naturally-infected durum wheat *cvs* with different degrees of susceptibility to FHB over 8 growing seasons, in NW Italy. Previously, only a few studies had reported the co-occurrence of the aforementioned mycotoxins and these studies always referred to just a few years, mainly in areas of Central and Southern Italy [18,22,28], France [29], Poland [30], Canada [31] and Argentina [32], without comparing *cvs* with different degrees of susceptibility to FHB under field conditions in order to evaluate their contamination by emerging mycotoxins. In such a context, it is important to underline that the recent increase in demand for pasta products has also led to an expansion of the growth of winter durum wheat to non-traditional growing areas (such as Northern Italy, Austria, Germany and France), where there are more humid climatic conditions [4,28,33]. These factors have resulted in a higher risk of FHB infections, and, consequently, of mycotoxin contamination, thereby limiting their potential cultivation in these areas [4,28,33].

In the present study, which was specifically carried out in a location highly susceptible to FHB to allow an effective varietal screening, almost all the detected mycotoxins showed higher levels of contamination than those reported for trials conducted in Central and Southern Italy in the same years [18,28]. In relation to this aspect, the data highlight that the growing seasons characterized by frequent and heavy rainfall (>120 mm), from wheat heading to the end of flowering (May), such as the 2016 and 2018 growing seasons, were also the years with the highest contamination of DON, ZEA, MON and ENNs.

Moreover, in the last few years, FHB has significantly increased on small-grain cereals, due to changes in crop management practices, minimum or reduced tillage, the intensification of maize in crop rotation and to weather patterns with more humidity and warm temperatures during anthesis as a result of climate change effects [34,35].

Like DON, the most susceptible *cv,* that is, Saragolla, exceeded the maximum level fixed for ZEA by EC Regulation No. 1881/2006 (100 µg kg^−1^) [12]. in 2014, 2016, 2018 and 2020, reaching the highest levels of contamination of 325 and 341 µg kg^−1^, respectively, in 2016 and 2020. ZEA is mainly produced by *F. graminearum* and the related species (i.e., *F. culmorum*) in cereals [36]. Therefore, its presence is commonly related to DON production [37]. ZEA has been shown to cause reproductive disorders in laboratory animals. Although the toxicity of ZEA in humans has not been conclusively established, the limited evidence available would seem to indicate that ZEA can cause the hyper estrogenic syndrome [38].

As far as the sanitary risk of modified and emerging mycotoxins is concerned, DON-3-G represents an additional risk to DON for human and animal health. In fact, this associated form could be hydrolyzed in the digestive tract of mammals, thereby contributing to the total dietary exposure of individuals to DON [39]. For this reason, together with its native form and with 3-ADON and 15-ADON, it should also be taken into account for correct risk assessments and food safety [40,41,42]. Nevertheless, to date, no regulatory limits have been established concerning the presence of MON and ENNs. Jonsson et al. [43] reported a high acute toxicity of MON in rats, with the LD_50_ value being at the same level as that of T-2 and HT-2 toxins, the most toxic of the *Fusarium* mycotoxins. In addition, Prosperini et al. [44] reported that, although in the last decade novel findings about a potential therapeutic action of ENN B have been presented, several in vitro and in vivo studies have revealed that ENN B interacts with primary target molecules, affects the biological response of cell defenses, promotes cell damage and produces potential interactions between food contaminants (particularly other mycotoxins), thus leading to abnormally high responses and to other molecular events underlying ENN B toxicity.

The development of more tolerant varieties is currently the most effective approach for controlling FHB and the occurrence of mycotoxins. The present data clearly show that the *cvs* with lower FHB susceptibility resulted in the lowest occurrence of DON, but also its modified forms (DON-3-G), ZEA and emerging mycotoxins, such as MON and ENNs.

Furthermore, in agreement with Berthiller et al., Dall’Asta et al. and Lemmens et al. [45,46,47], the most FHB resistant lines showed relatively more DON glycosylated (up to 30%) than susceptible *cvs*. In fact, the present data point out that the amount of DON-3-G relative to DON contamination varied between 5 and 15% and was influenced by both the genotype and the environment. Lemmens et al. [47] pointed out that since introgressing FHB resistance reduces both DON and DON-3-G levels in grain, although this reduction is lower for masked toxins, the specific resistance QTL (e.g., *Fhb1*) may enhance the speed or rate of DON detoxification. However, variances regarding the DON-3-G content, compared to its native form, and its distribution trend could also be explained by considering the different metabolic properties of each genotype in relation to that of fungi [48]. In addition to the genotype, the environment and the meteorological trend during ripening also play fundamental roles in establishing the extent and the relative amount of the contamination by co-occurring mycotoxins. In fact, depending on when a plant goes into the senescence period, its metabolism can be almost deactivated, thus preventing it from producing the glucoside form of DON [48]. In agreement with this statement, the highest DON-3-G/DON ratios were here recorded in 2013 and 2014, and these are also the years that were characterized by the longest ripening periods, as shown by the later harvesting time than for the other years, as a consequence of well distributed rainfall during ripening. Furthermore, the fungi, which developed to a greater extent on the peripheral tissues of the caryopsis, were still able to produce mycotoxins, as long as the moisture content during the dry-down process remained above 20%.

As far as the genotype × environment interaction is concerned, regardless of the pressure of the FHB disease, an inverse relationship between the level of contamination of DON and the DON-3-G/DON ratio was always recorded for all the *cvs* and years. This negative correlation was in agreement with studies carried out by Dall’Asta et al. [46], Ovando-Martínez et al., Audenaert et al. and Schweiger et al. [49,50,51]. These data suggest that the plant process of DON conjugation to glucose occurs with an intensity that is barely influenced by the occurrence of the free mycotoxin form, but which instead appears closely connected to the metabolic rate and the threshold biosynthetic ability of each *cv*. Thus, when the conditions (meteorological trend in spring and *cv* susceptibility) favor a high and quick increase in DON content during ripening, the DON-3-G/DON ratio tends to decrease, mainly because of high denominator values and an unequal proportional conversion to DON-3-G.

The absolute levels of the emerging mycotoxin contaminations highlight that after DON and its modified forms, ENNs represent the mycotoxin group with the highest content. As confirmed from the collected data, MON and ENNs are present at remarkably high contamination levels, in particular when adverse meteorological events occurred. Focusing in the years with the highest FHB severity, 2016 resulted in the highest contamination of MON and ENNs, while 2018 reported the highest DON content. Both years showed similar and high rainfall in May, while the temperatures from flowering to the ripening (May–June) were clearly lower in 2016 compared to 2018.

In 2016, and considering the most susceptible *cv* Saragolla, the MON and ENNS were 747 and 4014 µg kg^−1^, respectively.

To the best of the authors’ knowledge, the relative ratios between the emerging MON and ENN mycotoxins and the target mycotoxins of the FHB in wheat, DON, have never been reported before. These ratios indirectly indicate the variable resistance of the compared genotypes to the infections by the different *Fusarium* species that are able to produce their own characteristic mycotoxins. As already reported in the literature, although DON is mainly produced by *F. graminearum* and *F. culmorum* [11], *F. avenaceum* is the *Fusarium* species that is most able to biosynthesize depsipetides, such as ENN analogues and MON [18]. Like the DON-3-G/DON ratio, the MON/DON ratio confirms that, although the absolute MON levels of contamination for the more tolerant *cvs* were always the lowest, they showed higher MON biosynthetic ability than the susceptible ones, probably as a consequence of being relatively more prone to *F. avenaceum* infection. This ratio remained constantly higher in the FHB tolerant *cvs* in the years with both high (2016) and low (2017) DON contamination. Conversely, the ENN TOT/DON TOT ratio showed more variable behavior, as a function of the different years, and a less intense relationship with *cv* tolerance to FHB. This behavior could be linked to either a greater relative infection of *F. avenaceum* than *F. graminearum,* or a greater toxigenesis in relation to the environmental conditions that could favor *F. avenaceum vs F. graminearum* in different ways. However, only considering the presence of mycotoxins, it could be assumed that there is no real effect of flora inversion [52], because DON remained the main toxin, and the more tolerant *cvs* were also the least contaminated by all the other mycotoxins, such as the MON and ENNs mainly produced by *F. avenaceum*.

In conclusion, the present data underline that, even though the relative relationships between modified/emerging mycotoxins and native/target mycotoxins (DON, or DON TOT) could be higher in the more tolerant *cvs* than in susceptible ones, the genotypes that have a lower FHB susceptibility and/or DON biosynthetic ability also show lower absolute levels of contamination by other emerging and modified mycotoxins. In accordance with the consumers’ and supply chains’ request to reduce the use of pesticides, and as promoted in public policies (e.g., the *farm-to-fork strategy* in the EU), the selection of wheat genotypes resistant to FHB seems to be the most effective approach for controlling the overall mycotoxin risk, as confirmed in this experiment over several growing seasons with different meteorological trends.

Nevertheless, the most tolerant *cvs* that are also able to combine other food chain requirements, such as an adequate yield capacity, a high protein content, compliance with other qualitative parameters and a low susceptibility to other diseases (e.g., septoria leaf blotch), should be inserted into cropping system and breeding programs designed to control not only the native mycotoxins, but also their modified forms and the emerging mycotoxins.

## Figures and Tables

**Figure 1 microorganisms-09-00408-f001:**
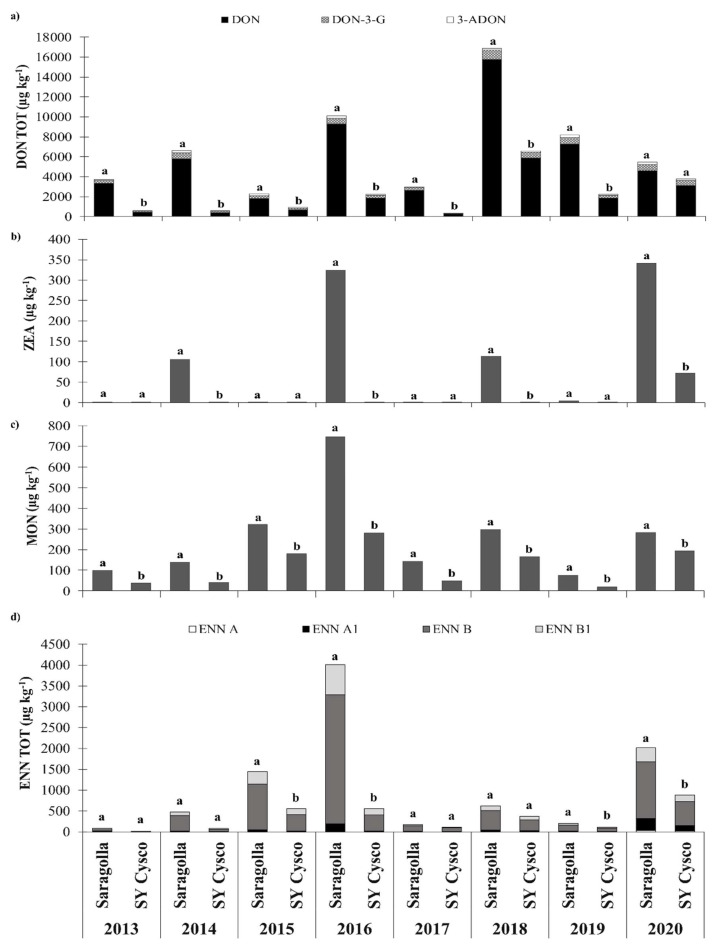
Effects of the growing season and durum wheat cultivar with different degrees of FHB susceptibility on the mycotoxin content in NW Italy during the 2012–2020 period: (**a**) Total deoxynivalenol content (DON TOT = sum of deoxynivalenol, DON; deoxynivalenol-3-glucoside, DON-3-G; and 3-acetyldeoxynivalenol, 3-ADON); (**b**) Zearalenone (ZEA) content; (**c**) Moniliformin (MON) content; (**d**) Total enniatin content (ENN TOT = sum of enniatin A, ENN A; enniatin A_1_, ENN A_1_; enniatin B, ENN B; and enniatin B_1_, ENN B_1_). Different letters above the bars indicate significant differences between cultivars for each growing season (*p* < 0.05). The reported data are the average of 3 replications. Statistical analysis was performed on the transformed [T; y’= ln (x + 1)] mycotoxin concentration values.

**Figure 2 microorganisms-09-00408-f002:**
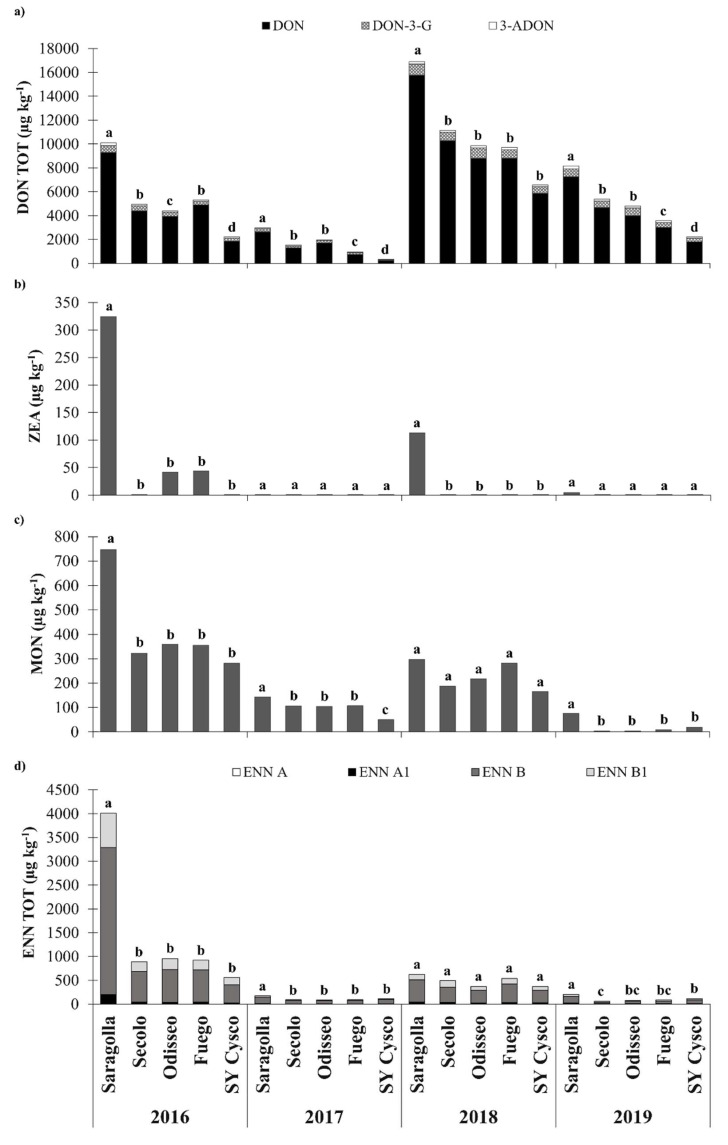
Effects of the growing season and durum wheat cultivar with different degrees of FHB susceptibility on the mycotoxin content in NW Italy during the 2015–2019 period: (**a**) Total deoxynivalenol content (DON TOT = sum of deoxynivalenol, DON; deoxynivalenol-3-glucoside, DON-3-G; and 3-acetyldeoxynivalenol, 3-ADON); (**b**) Zearalenone (ZEA) content; (**c**) Moniliformin (MON) content; (**d**) Total enniatin content (ENN TOT = sum of enniatin A, ENN A; enniatin A_1_, ENN A_1_; enniatin B, ENN B; and enniatin B_1_, ENN B_1_). Different letters above the bars indicate significant differences between cultivars for each growing season (*p* < 0.05). The reported data are the average of 3 replications.

**Figure 3 microorganisms-09-00408-f003:**
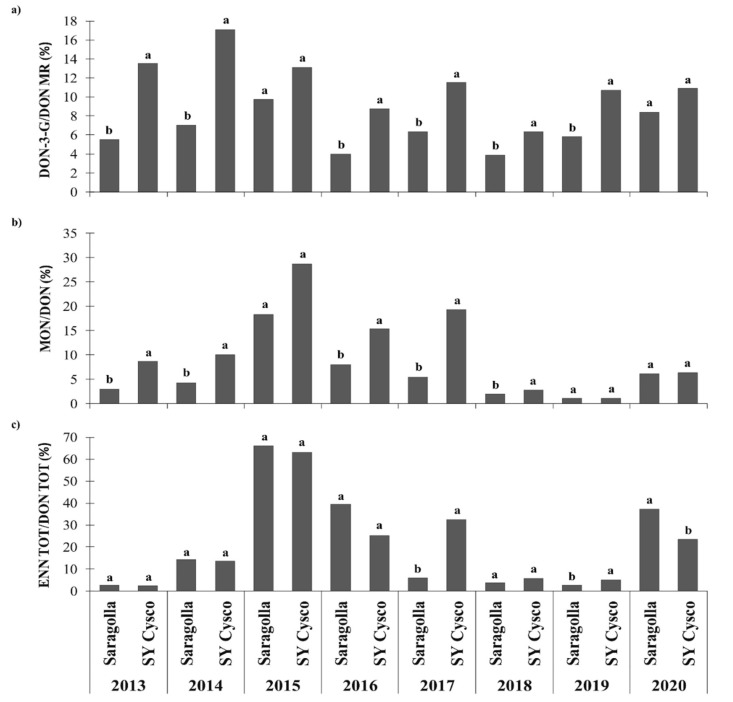
Effects of the growing season and durum wheat cultivar with different degrees of FHB susceptibility on the mycotoxin content ratios in NW Italy during the 2012–2020 period: (**a**) Deoxynivalenol-3-glucoside/deoxynivalenol molar ratio (DON-3-G/DON MR); (**b**) Moniliformin/Deoxynivalenol content ratio (MON/DON); (**c**) Total enniatin/Total deoxynivalenol content ratio (ENN TOT/DON TOT: ENN TOT = sum of enniatin A, ENN A; enniatin A_1_, ENN A_1_; enniatin B, ENN B; and enniatin B_1_, ENN B_1_; DON TOT = sum of deoxynivalenol, DON; deoxynivalenol-3-glucoside, DON-3-G; and 3-acetyldeoxynivalenol, 3-ADON). Different letters above the bars indicate significant differences between cultivars for each growing season (*p* < 0.05). The reported data are the average of 3 replications.

**Figure 4 microorganisms-09-00408-f004:**
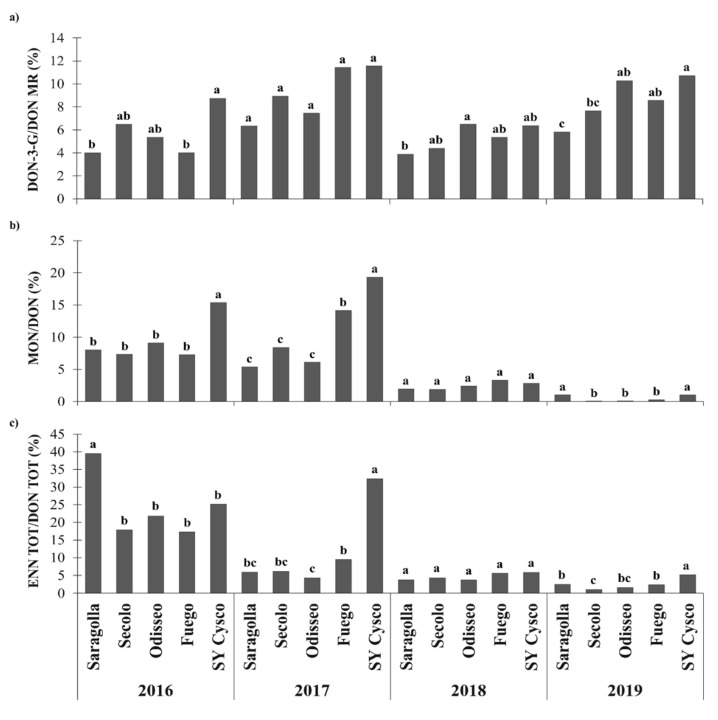
Effects of the growing season and durum wheat cultivar with different degrees of FHB susceptibility on the mycotoxin content ratios in NW Italy during the 2015–2019 period: (**a**) Deoxynivalenol-3-glucoside/deoxynivalenol molar ratio (DON-3-G/DON MR); (**b**) Moniliformin/Deoxynivalenol content ratio (MON/DON); (**c**) Total enniatin/Total deoxynivalenol content ratio (ENN TOT/DON TOT: ENN TOT = sum of enniatin A, ENN A; enniatin A_1_, ENN A_1_; enniatin B, ENN B; and enniatin B_1_, ENN B_1_; DON TOT = sum of deoxynivalenol, DON; deoxynivalenol-3-glucoside, DON-3-G; and 3-acetyldeoxynivalenol, 3-ADON). Different letters above the bars indicate significant differences between cultivars for each growing season (*p* < 0.05). The reported data are the average of 3 replications.

**Table 1 microorganisms-09-00408-t001:** Main trial information for the field experiments conducted in Cigliano (NW Italy) in the 2012–2020 period.

Year	Sowing Date	N Fertilization		Anthesis		Harvest Date	
		GS 23 ^1^	GS 51 ^1^		GS 65 ^1^		
2013	6 November 2012	11 March 2013	9 May 2013		17 May 2013	10 July 2013	
2014	27 October 2013	7 March 2014	23 April 2014		5 May 2014	10 July 2014	
2015	8 November 2014	12 March 2015	4 May 2015		10 May 2015	29 June 2015	
2016	6 November 2015	23 February 2016	3 May 2016		10 May 2016	1 July 2016	
2017	4 November 2016	7 March 2017	28 April 2017		12 May 2017	27 June 2017	
2018	31 October 2017	23 February 2018	7 May 2018		11 May 2018	3 July 2018	
2019	16 November 2018	6 March 2019	30 April 2019		16 May 2019	1 July 2019	
2020	6 November 2019	5 March 2020	23 April 2020		12 May 2020	29 June 2020	

^1^ Growth stage [24].

**Table 2 microorganisms-09-00408-t002:** Monthly rainfall and growing degree days (GDD) for each growing season compared to data from 1990–2010.

Parameters	Month	Average 1990–2010	2012–13	2013–14	2014–15	2015–16	2016–17	2017–18	2018–19	2019–20
Rainfall	November	85	182	68	438	5	158	48	124	314
(mm)	December	48	11	139	89	4	45	33	11	132
	January	38	17	117	36	14	4	107	6	5
	February	35	40	129	75	120	45	60	43	1
	March	38	118	71	88	52	69	109	17	62
	April	88	165	138	75	37	34	93	116	81
	May	105	101	84	96	171	79	188	178	122
	June	66	16	144	86	83	149	35	40	113
	Nov—June	503	650	890	982	486	582	673	535	830
GDD ^1^	November	205	272	251	303	270	238	224	292	249
(Σ °C d^−1^)	December	110	113	169	184	164	144	113	151	193
	January	99	142	149	159	119	97	178	141	168
	February	142	110	179	143	158	152	113	195	229
	March	240	208	335	304	252	349	223	314	285
	April	337	398	428	409	405	415	456	393	414
	May	510	480	515	569	490	554	583	478	579
	June	594	620	637	659	624	673	665	667	624
	Nov—June	2239	2343	2662	2728	2481	2622	2554	2632	2741

^1^ Accumulated growing degree days for each month using a 0 °C base value.

**Table 3 microorganisms-09-00408-t003:** Effects of the growing season and durum wheat cultivar with different degrees of *Fusarium* head blight (FHB) susceptibility on the grain yield (Yield), test weight (TW), FHB incidence (FHB Inc) and severity (FHB Sev) in NW Italy during the 2012—2020 period.

Factor	Source of Variation	Grain Yield	TW	FHB Incidence ^1^	FHB Severity ^2^
		(t ha^−1^)	(kg hL^−1^)	(%)	(%)
Year	2013	5.9 cd	80.8 a	47.5 cd	4.6 cd
	2014	7.3 a	77.8 b	28.6 e	2.0 d
	2015	4.3 f	80.5 a	39.3 de	1.4 d
	2016	6.5 bc	76.4 c	73.6 a	20.9 a
	2017	6.1 bc	77.8 b	46.2 cd	3.8 cd
	2018	5.0 e	74.8 d	44.7 cd	10.2 b
	2019	5.3 de	79.7 a	53.6 bc	6.7 bc
	2020	6.7 ab	79.3 ab	63.9 ab	6.7 bc
	*p*-value ^3^	<0.001	<0.001	<0.001	<0.001
	sem ^4^	2.6	5.5	36.3	16.6
Cultivar (*cv*)	Saragolla	5.6 b	76.2 b	71.3 a	12.4 a
	SY Cysco	6.0 a	80.6 a	27.2 b	1.7 b
	*p*-value ^3^	0.015	<0.001	<0.001	<0.001
	sem ^4^	0.2	3.0	30.9	7.5
Year × *cv*	*p*-value ^3^	0.006	<0.001	0.071	<0.001
	sem ^4^	1.3	3.9	18.1	18.9

The reported data are the average of 3 replications. ^1^ FHB incidence was calculated as the percentage of ears with FHB damage, considering 200 ears per sample. ^2^ FHB severity was calculated as the percentage of kernels per ear with FHB damage, considering 200 ears per sample. ^3^ Means followed by different letters are significantly different; the level of significance of ANOVA (*p*-value) is reported in the table. ^4^ sem = standard error of the means.

**Table 4 microorganisms-09-00408-t004:** Effects of the growing season and durum wheat cultivar, with different degrees of FHB susceptibility, on the grain yield (Yield), test weight (TW), FHB incidence and severity in NW Italy during the 2015–2019 period.

Factor	Source of Variation	Grain Yield	TW	FHB Incidence ^1^	FHB Severity ^2^
		(t ha^−1^)	(kg hL^−1^)	(%)	(%)
Year	2016	6.3 a	76.3 c	70.7 a	26.4 a
	2017	5.8 b	77.3 b	40.2 b	4.3 c
	2018	5.0 c	76.3 c	41.7 b	8.4 b
	2019	5.2 c	80.3 a	42.5 b	4.6 c
	*p*-value ^3^	<0.001	<0.001	<0.001	<0.001
	sem ^4^	1.0	3.5	26.4	18.3
Cultivar (*cv*)	Saragolla	5.3 b	74.5 d	77.9 a	17.9 b
	Fuego	5.3 b	78.1 bc	38.9 cd	3.8 cd
	Odisseo	5.5 b	77.4 c	42.4 c	6.4 c
	Secolo	5.6 b	78.1 bc	52.1 b	24.9 a
	SY Cysco	6.1 a	80.3 a	31.1 d	2.4 d
	*p*-value ^3^	<0.001	<0.001	<0.001	<0.001
	sem ^4^	0.6	4.3	37.8	18.3
Year × *cv*	*p*-value ^3^	0.029	<0.001	0.001	<0.001
	sem ^4^	1.3	3.3	35.4	47.3

The reported data are the average of 3 replications. ^1^ FHB incidence was calculated as the percentage of ears with FHB damage, considering 200 ears per sample. ^2^ FHB severity was calculated as the percentage of kernels per ear with FHB damage, considering 200 ears per sample. ^3^ Means followed by different letters are significantly different; the level of significance of ANOVA (*p*-value) is reported in the table. ^4^ sem = standard error of the means.

**Table 5 microorganisms-09-00408-t005:** Correlations between agro-meteorological parameters and mycotoxins.

Correlation	Rainfall May	Rainfall June	GDD Nov–June	GDD May	GDD June	Grain Yield	DON	DON-3-G	DON TOT	ZEA	MON	ENN TOT	DON-3-G/ DON MR	MON/ DON R	ENN TOT/ DON TOT R
Rainfall Nov–June	−**0.453** **	0.193	**0.501** **	**0.530** **	−0.021	−0.061	-0.094	−0.027	−0.088	0.056	−0.061	0.018	**0.375** **	**0.390** **	**0.485** **
Rainfall May		−**0.740** **	−**0.244** *	−0.203	−0.006	−**0.258** *	**0.612** **	**0.620** **	**0.619** **	0.066	**0.244 ***	0.167	−**0.480** **	−**0.519** **	−**0.275** *
Rainfall June			**0.435** **	**0.259** *	0.057	**0.424** **	−**0.435** **	−**0.451** **	−**0.440** **	0.114	0.032	0.048	**0.278** *	**0.449** **	**0.288** **
GDD Nov–June				**0.384** **	**0.516** **	−0.135	−0.163	0.020	−0.147	0.030	−**0.247** *	−0.063	**0.322** **	0.180	**0.317** **
GDD May					**0.384** **	−**0.262** *	**0.232** *	0.186	**0.228** *	0.041	0.134	0.027	−0.041	**0.238** *	**0.219** *
GDD June						−**0.537** **	0.059	0.108	0.061	−**0.381** **	−**0.450** **	−**0.427** **	0.038	−0.115	−**0.230** *
Grain Yield							−**0.231** *	−0.192	−**0.230** *	**0.282** **	0.063	0.130	0.206	−0.011	−0.035
DON								**0.870** **	**0.999** **	**0.378** **	**0.414** **	**0.295** **	−**0.646** **	−**0.483** **	−**0.245** *
DON-3-G									**0.889** **	**0.298** **	**0.223** *	0.177	−**0.457** **	−**0.569** **	−**0.294** **
DON TOT										**0.380** **	**0.406** **	**0.292** **	−**0.637** **	−**0.493** **	−**0.247** *
ZEA											**0.589** **	**0.728** **	−**0.245** *	−0.048	**0.265** *
MON												**0.884** **	−**0.468** **	0.204	**0.485** **
ENN TOT													−**0.307** **	0.160	**0.560** **
DON-3-G/DON MR														**0.427** **	0.171
MON/DON R															**0.800** **

The data reported in the table are Pearson product-moment correlation coefficients. (*) = correlation significant at *p* ≤ 0.05; (**) correlation significant at *p* ≤ 0.01. Growing degree days (GDD), Deoxynivalenol (DON), deoxynivalenol-3-glucoside (DON-3-G), total deoxynivalenol content (DON TOT = sum of DON, DON-3-G and 3-acetyldeoxynivalenol (3-ADON)), zearalenone (ZEA), moniliformin (MON), total enniatin content (ENN TOT = sum of enniatin A (ENN A), enniatin A_1_ (ENN A_1_), enniatin B (ENN B) and enniatin B_1_ (ENN B_1_)), deoxynivalenol-3-glucoside/deoxynivalenol molar ratio (DON-3-G/DON MR), moniliformin/deoxynivalenol content ratio (MON/DON R), total enniatin/total deoxynivalenol content ratio (ENN TOT/DON TOT R).

## Data Availability

The datasets generated analyzed during the current study are available from the corresponding author on reasonable request.

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
