# Peer review of "Effects of Durum Wheat Cultivars with Different Degrees of FHB Susceptibility Grown under Different Meteorological Conditions on the Contamination of Regulated, Modified and Emerging Mycotoxins"

_microorganisms, 2021, doi:10.3390/microorganisms9020408_

Round 1

Reviewer 1 Report

This manuscript is well written and pleasant to read. The purpose of the manuscript is important in human health and plant selection. I have only a few minor suggestions to make.

L29-66: In my opinion this section is too long and should be synthesized

L81-106: please revise to highlight which toxins are expected on durum wheat cvs. I suppose aflatoxins, ergot alkaloids are not expected, but what about zearalenone?

L148-150 and L166: please clarify how it was checked that 5g of flour are representative of 4kg of wheat

Table2-4 & Fig 1-2: Is it possible to correlate some of the variables measured?

L288-300, Figure 1-2: Please use different color/symbol for ENN, what was shown in actual fig 1-2 is very difficult to read. Please clarify in text whether the relative rate of each ENN varied with year/cultivar or not.

L425-434: Can you add some references that support the important role of the cultivar in the glycosylation rate?

Author Response

This manuscript is well written and pleasant to read. The purpose of the manuscript is important in human health and plant selection.

We would like to thank the reviewer for appreciating the manuscript.

I have only a few minor suggestions to make:

  • L29-66: In my opinion this section is too long and should be synthesized.

As suggested by the reviewer the section has been synthesized, deleting the following sentences:

  • (P1, L31-34): “Certain regions in the Mediterranean Basin and North America (i.e. Canada) produce about 60% of the world’s durum wheat, which is mainly used for human consumption as pasta and other food products, such as bulgur, couscous and some types of bread.”
  • (P2, L50-52): “Therefore, wheat producers need to face a variety of factors, including abiotic and bio-tic stresses, which could influence both yields and quality.”

  • L81-106: please revise to highlight which toxins are expected on durum wheat cvs. I suppose aflatoxins, ergot alkaloids are not expected, but what about zearalenone?

In order to highlight which toxins are expected on durum wheat the following sentence has been added (P2, L86-88): “Among the regulated mycotoxins, in addition to DON, also T-2 and HT-2 toxins, zearalenone and ochratoxin A (OTA) could be found to contaminate durum wheat in field, while the occurrence of aflatoxins (AFs) are infrequent [18].”.

  • L148-150 and L166: please clarify how it was checked that 5g of flour are representative of 4kg of wheat.

5g of each flour are representative of 4kg of each wheat grain because the whole amount of grain was ground, mixed thoroughly and then a sub-sample of 5g of flour was taken and subjected to analysis to assess the mycotoxin content. In order to clarify this aspect, the sentence at P4, L161-164 has been modified as follow: “The harvested grains were mixed thoroughly, and 4 kg grain samples were taken from each plot, ground using a ZM 200 Ultra Centrifugal Mill (Retsch GmbH, Haan, Ger-many) and representative sub-samples of each flour were used directly to analyze the mycotoxin content.”.

  • Table2-4 & Fig 1-2: Is it possible to correlate some of the variables measured?

As requested by the reviewer a Table (Table 5) and a new paragraph named “3.6. Correlations between agro-meteorological parameters and mycotoxins” (P13, L399-409) have been added in order to investigate the possible correlations between some of the variables measured. Moreover, for this purpose the following sentences have been added at:

  • P5, L219-222: “Simple correlation coefficients were obtained for mycotoxins, relative to each other and to meteorological (rainfall and GDD referred to the period November-June, May and June) and agronomic (grain yield and FHB severity) parameters, by joining the data sets that referred to the eight growing seasons.”
  • P13, L399-409: “Grain yield significantly and negatively correlated with the rainfall occurred in May and the GDD recorded in May and June, while was positively correlated with the rainfall occurred in June (Table 5). As far as the correlation among mycotoxins, relative to each other and to agro-meteorological parameters was concerned, the occurrence of DON was closely related to the rainfall occurred in May and although less intensely it correlated significantly with ZEA, MON and ENN TOT. DON-3-G exhibited DON-like behavior, with which it was closely related. On the other hand, the DON-3-G/DON ratio was negatively related to the DON content. ZEA, MON and ENN TOT were closely related to each other, and negatively related to the GDD recorded in June, while, except for MON, they were not influenced by the rainfall occurred in May.”

  • L288-300, Figure 1-2: Please use different color/symbol for ENN, what was shown in actual fig 1-2 is very difficult to read. Please clarify in text whether the relative rate of each ENN varied with year/cultivar or not.

As suggested by the reviewer the color for ENN were changed in Fig. 1 and Fig. 2. Moreover, in order to clarify in text whether the relative rate of each ENN varied with year/cultivar or not the following sentence has been added at P9, L336-340: “On average ENN A, ENN A1, ENN B, ENN B1 accounted respectively for 1%, 7%, 71% and 21% of the total enniatin contamination level. The relative rate of each ENN form was quite stable among cvs and it varied on average over the years within about ±10-15% of the previous mentioned values.”

  • L425-434: Can you add some references that support the important role of the cultivar in the glycosylation rate?

As suggested by the reviewer 3 supporting references have been added (Ovando-Martínez et al.,2013; Audenaert et al., 2013 and Schweiger et al., 2013 [49-51]) and the sentence at P16, L492-496 has been modified as follow: “As far as the genotype × environment interaction is concerned, regardless of the pressure of the FHB disease, an inverse relationship between the level of contamination of DON and the DON-3-G/DON ratio was always recorded for all the cvs and years. This negative correlation was in agreement with studies carried out by Dall’Asta et al. [46], Ovando-Martínez et al., Audenaert et al. and Schweiger et al. [49-51].”

Reviewer 2 Report

Lines 127-130 The experiment design, planting fields, fungicide application and planting dates for Secolo, Odisseo and Fuego cv should be given in more details.

Lines 221-222 “A greater difference was recorded for the yield parameters between SY Cysco and Saragolla in 2016, 2017 and 2019.” → This statement cannot be verified by looking at the content of Table 3.

Line 233 “FHB incidence and severity were higher in 2016, followed by 2018, 2019 and 2020” → “FHB incidence and severity were higher in 2016, followed by 2020, 2019 and 2018”

Lines 240-241 “The differences between 240 susceptible cv Saragolla and the other varieties were higher in 2016, 2018 and 2019. → This statement cannot be verified by looking at the content of Table 4.

Lines 254-256 “The content of this mycotoxin was clearly related to the meteorological conditions, particularly close to anthesis, in each growing season.” → This statement cannot be verified by looking at the content of Figure 1.

Lines 266-268 “a moderately tolerant cv, SY Cysco, allowed the limits to be complied with, without the need of any fungicide application, for half of the considered growing seasons” → It is not the case for SY Cysco in 2018 and 2020.

Line 298 “Sy Cysco” →“SY Cysco”

Lines 369-371 “the growing seasons characterized by frequent and heavy rainfall (> 120 mm), from wheat heading to the end of flowering (May), such as the 2016 and 2018 growing seasons, were also the years with the highest contamination of DON, ZEA, MON and ENNs.” → Please discuss why 2017 had heavy rainfall, but low contamination of DON, ZEA, MON and ENNs.

Lines 452-460 Please add discussion of difference between 2016 and 2018 in terms of rainfall, GDD, FHB incidence, FHB severity, DON TOT, ENT TOT, and DON TOT/ ENT TOT ratio.  FHB resistant cv may not solve or prevent the problem of regulated and emerging mycotoxins in the same time.

Author Response

  • Lines 127-130 The experiment design, planting fields, fungicide application and planting dates for Secolo, Odisseo and Fuego cv should be given in more details.

Secolo, Odisseo and Fuego cvs were compared in the same experimental field of SY Cysco and Saragolla cvs and thus all the details requested by the reviewer are already present through the paragraph “2.1 Design of the field experiment and samples”. Nevertheless, the sentence at P3, L138-141 has been modified as follow: “Secolo, Odisseo and Fuego cvs, which are characterized by an intermediate susceptibility to DON contamination, were also compared in the same experimental field, albeit only over four years (2016-2019). All the cvs were provided by Syngenta Italia Spa Milano, Italy.”

  • Lines 221-222 “A greater difference was recorded for the yield parameters between SY Cysco and Saragolla in 2016, 2017 and 2019.” → This statement cannot be verified by looking at the content of Table 3.

In order to support with data the statement highlighted by the reviewer the sentence at P7, L260-265 has been modified as follow: “A greater difference was recorded for the yield parameters between SY Cysco and Saragolla in 2016 (grain yield values of 6.7 and 6.3 t ha-1 and TW values of 79.8 and 73.0 kg hL-1 for SY Cysco and Saragolla, respectively), 2017 (grain yield values of 6.4 and 5.8 t ha-1 and TW values of 80.4 and 75.3 kg hL-1 for SY Cysco and Saragolla, respectively) and 2019 (grain yield values of 6.1 and 4.4 t ha-1 and TW values of 82.5 and 77.95 kg hL-1 for SY Cysco and Saragolla, respectively).”

  • Line 233 “FHB incidence and severity were higher in 2016, followed by 2018, 2019 and 2020” → “FHB incidence and severity were higher in 2016, followed by 2020, 2019 and 2018”.

The sentence at P7, L275-276 has been modified as suggested by the reviewer.

  • Lines 240-241 “The differences between susceptible cv Saragolla and the other varieties were higher in 2016, 2018 and 2019. → This statement cannot be verified by looking at the content of Table 4.

In order to support with data the statement highlighted by the reviewer, the sentence at P7, L283-288 has been modified as follow: “The differences between susceptible cv Saragolla and SY Cysco were higher in 2016 (FHB incidence values of 98.7 and 48.4% and FHB severity values of 36.7 and 5.0% for Saragolla and SY Cysco, respectively), 2018 (FHB incidence values of 71.3 and 18.1% and FHB se-verity values of 19.8 and 0.6% for Saragolla and SY Cysco, respectively) and 2019 (FHB in-cidence values of 75.0 and 32.2% and FHB severity values of 11.7 and 1.8 % for Saragolla and SY Cysco, respectively).”

  • Lines 254-256 “The content of this mycotoxin was clearly related to the meteorological conditions, particularly close to anthesis, in each growing season.” → This statement cannot be verified by looking at the content of Figure 1.

In order to clarify the relation among the mycotoxins and their interaction with the agro-meteorological parameters, a correlation table (Table 5) and a new paragraph named “3.6. Correlations between agro-meteorological parameters and mycotoxins” (P13, L399-409) have been added.

  • Lines 266-268 “a moderately tolerant cv, SY Cysco, allowed the limits to be complied with, without the need of any fungicide application, for half of the considered growing seasons” → It is not the case for SY Cysco in 2018 and 2020.

As suggested by the reviewer the sentence at P8, L312-315 has been modified as follow: “On the other hand, the cultivation of a moderately tolerant cv, SY Cysco, allowed the limits to be complied with, without the need of any fungicide application, for half of the considered growing seasons, except in 2016, 2018, 2019 and 2020.”

  • Line 298 “Sy Cysco” →“SY Cysco”.

The name of the cv has been modified at P9, L348, as suggested by the reviewer.

  • Lines 369-371 “the growing seasons characterized by frequent and heavy rainfall (> 120 mm), from wheat heading to the end of flowering (May), such as the 2016 and 2018 growing seasons, were also the years with the highest contamination of DON, ZEA, MON and ENNs.” → Please discuss why 2017 had heavy rainfall, but low contamination of DON, ZEA, MON and ENNs.

Looking at Table 2 it is possible to see that in May 2017 were recorded only 79 mm of rainfall. Moreover, May 2017 was also the lowest rainy month compared to the same month of all the other growing seasons. Rainfall occur in 2017 in June, but too late to influence the development of the FHB.

  • Lines 452-460 Please add discussion of difference between 2016 and 2018 in terms of rainfall, GDD, FHB incidence, FHB severity, DON TOT, ENT TOT, and DON TOT/ ENT TOT ratio. FHB resistant cv may not solve or prevent the problem of regulated and emerging mycotoxins in the same time.

As suggested by the reviewer, in order to discuss the difference between 2016 and 2018 in terms of meteorological and mycotoxin contamination, the following sentence has been added at P16, L508-511: “Focusing in the years with the highest FHB severity, 2016 resulted in the highest contamination of MON and ENNs, while 2018 reported the highest DON content. Both years showed similar and high rainfall in May, while the temperatures from flowering to the ripening (May – June) were clearly lower in 2016 compared to 2018.”